# ProxSGD: Training Structured Neural Networks under Regularization and Constraints

**Yang Yang**
Fraunhofer ITWM
Fraunhofer Center Machine Learning
yang.yang@itwm.fraunhofer.de

**Yaxiong Yuan**
University of Luxembourg
yaxiong.yuan@uni.lu

**Avraam Chatzimichailidis**
Fraunhofer ITWM
TU Kaiserslautern
avraam.chatzimichailidis@itwm.fraunhofer.de

**Ruud JG van Sloun**
Eindhoven University of Technology
r.j.g.v.sloun@tue.nl

**Lei Lei, Symeon Chatzinotas**
University of Luxembourg
{lei.lei, symeon.chatzinotas}@uni.lu

## Abstract

In this paper, we consider the problem of training structured neural networks (NN) with nonsmooth regularization (e.g. $\ell_1$-norm) and constraints (e.g. interval constraints). We formulate training as a constrained nonsmooth nonconvex optimization problem, and propose a convergent proximal-type stochastic gradient descent (ProxSGD) algorithm. We show that under properly selected learning rates, with probability 1, every limit point of the sequence generated by the proposed ProxSGD algorithm is a stationary point. Finally, to support the theoretical analysis and demonstrate the flexibility of ProxSGD, we show by extensive numerical tests how ProxSGD can be used to train either sparse or binary neural networks through an adequate selection of the regularization function and constraint set.

## 1 Introduction

In this paper, we consider the problem of training neural networks (NN) under constraints and regularization. It is formulated as an optimization problem

$$\underset{\boldsymbol{x} \in \mathbb{X} \subseteq \mathbb{R}^n}{\text{minimize}} \underbrace{\frac{1}{m} \sum_{i=1}^{m} f_i(\boldsymbol{x}, \boldsymbol{y}_i)}_{\triangleq f(\boldsymbol{x})} + r(\boldsymbol{x}), \tag{1}$$

where $\boldsymbol{x}$ is the parameter vector to optimize, $\boldsymbol{y}_i$ is the $i$-th training example which consists of the training input and desired output, and $m$ is the number of training examples. The training loss $f$ is assumed to be smooth (but nonconvex) with respect to $\boldsymbol{x}$, the regularization $r$ is assumed to be convex (but nonsmooth), proper and lower semicontinuous, and the constraint set $\mathbb{X}$ is convex and compact (closed and bounded).

When $r(\boldsymbol{x}) = 0$ and $\mathbb{X} = \mathbb{R}^n$, stochastic gradient descent (SGD) has been used to solve the optimization problem (1). At each iteration, a minibatch of the $m$ training examples are drawn randomly, and the obtained gradient is an unbiased estimate of the true gradient. Therefore SGD generally moves along the descent direction, see Bertsekas & Tsitsiklis (2000). SGD can be accelerated by replacing the instantaneous gradient estimates by a momentum aggregating all gradient in past iterations. Despite the success and popularity of SGD with momentum, its convergence had been an open problem. Assuming $f$ is convex, analyzing the convergence was first attempted in Kingma & Ba (2015) and later concluded in Reddi et al. (2018). The proof for a nonconvex $f$ was later given in Chen et al. (2019); Lei et al. (2019).

In machine learning, the regularization function $r$ is typically used to promote a certain structure in the optimal solution, for example sparsity as in, e.g., feature selection and compressed sensing, or a zero-mean-Gaussian prior on the parameters (Bach et al., 2011; Boyd et al., 2010). It can be interpreted as a penalty function since at the optimal point $\boldsymbol{x}^\star$ of problem (1), the value $r(\boldsymbol{x}^\star)$ will be small. One nominant example is the Tikhonov regularization $r(\boldsymbol{x}) = \mu\|\boldsymbol{x}\|_2^2$ for some predefined constant $\mu$, and it can be used to alleviate the ill-conditioning and ensure that the magnitude of the weights will not become exceedingly large. Another commonly used regularization, the $\ell_1$-norm where $r(\boldsymbol{x}) = \mu\|\boldsymbol{x}\|_1 = \mu\sum_{j=1}^n |x_j|$ (the convex surrogate of the $\ell_0$-norm), would encourage a sparse solution. In the context of NN, it is used to (i) promote a sparse neural network (SNN) to alleviate overfitting and to allow a better generalization, (ii) accelerate the training process, and (iii) prune the network to reduce its complexity, see Louizos et al. (2018) and Gale et al. (2019).

Technically, it is difficult to analyze the regularizations as some commonly used convex regularizers are nonsmooth, for example, $\ell_1$-norm. In current implementations of TensorFlow, the gradient of $|x|$ is simply set to 0 when $x = 0$. This amounts to the stochastic subgradient descent method and usually exhibits slow convergence. Other techniques to promote a SNN includes magnitude pruning and variational dropout, see Gale et al. (2019).

Although regularization can be interpreted as a constraint from the duality theory, sometimes it may still be more desirable to use explicit constraints, for example, $\sum x_j^2 \leq \alpha$, where the summation is over the weights on the same layer. This is useful when we already know how to choose $\alpha$. Another example is the lower and upper bound on the weights, that is, $\boldsymbol{l} \leq \boldsymbol{w} \leq \boldsymbol{u}$ for some predefined $\boldsymbol{l}$ and $\boldsymbol{u}$. Compared with regularization, constraints do not encourage the weights to stay in a small neighborhood of the initial weight, see Chapter 7.2 of Goodfellow et al. (2016) for more details.

The set $\mathbb{X}$ models such explicit constraints, but it poses an additional challenge for stochastic gradient algorithms as the new weight obtained from the SGD method (with or without momentum) must be projected back to the set $\mathbb{X}$ to maintain its feasibility. However, projection is a nonlinear operator, so the unbiasedness of the random gradient would be lost. Therefore the convergence analysis for constrained problems is much more involved than unconstrained problems.

In this paper, we propose a convergent proximal-type stochastic gradient algorithm (ProxSGD) to train neural networks under nonsmooth regularization and constraints. It turns out momentum plays a central role in the convergence analysis. We establish that with probability (w.p.) 1, every limit point of the sequence generated by ProxSGD is a stationary point of the nonsmooth nonconvex problem (1). This is in sharp contrast to unconstrained optimization, where the convergence of the vanilla SGD method has long been well understood while the convergence of the SGD method with momentum was only settled recently. Nevertheless, the convergence rate of ProxSGD is not derived in the current work and is worth further investigating.

To test the proposed algorithm, we consider two applications. The first application is to train a SNN, and we leverage $\ell_1$-regularization, that is,

$$\operatorname*{minimize}_{\boldsymbol{x}} \quad \frac{1}{m}\sum_{i=1}^m f_i(\boldsymbol{x}, \boldsymbol{y}_i) + \mu\|\boldsymbol{x}\|_1. \tag{2}$$

The second application is to train a binary neural network (BNN) where the weights (and activations) are either 1 or -1 (see Courbariaux et al. (2015; 2016); Hou et al. (2017); Yin et al. (2018); Bai et al. (2019) for more details). To achieve this, we augment the loss function with a term that penalizes the weights if they are not +1 or -1:

$$\operatorname*{minimize}_{\boldsymbol{x}, \boldsymbol{a}} \quad \frac{1}{m}\sum_{i=1}^m f_i(\boldsymbol{x}, \boldsymbol{y}_i) + \frac{\mu}{4}\sum_{j=1}^n (a_j(x_j+1)^2 + (1-a_j)(x_j-1)^2)$$

$$\text{subject to} \quad a_j = 0 \text{ or } 1, j = 1, \dots, n,$$

where $\mu$ is a given penalty parameter. The binary variable $a_j$ can be interpreted as a switch for weight $x_j$: when $a_j = 0$, $(1-a_j)(x_j-1)^2$ is activated, and there is a strong incentive for $x_j$ to be 1 (the analysis for $a_j = 1$ is similar). Since integer variables are difficult to optimize, we relax $a_j$ to be a continuous variable between 0 and 1. To summarize, a BNN can be obtained by solving the

following regularized optimization problem under constraints with respect to $\boldsymbol{x}$ and $\boldsymbol{a}$

$$\underset{\boldsymbol{x},\boldsymbol{a}}{\text{minimize}} \quad \frac{1}{m}\sum_{i=1}^{m} f_i(\boldsymbol{x}, \boldsymbol{y}_i) + \frac{\mu}{4}\sum_{j=1}^{n}(a_j(x_j+1)^2 + (1-a_j)(x_j-1)^2)$$

$$\text{subject to} \quad 0 \le a_j \le 1, j = 1, \ldots, n. \tag{3}$$

If $\mu$ is properly selected (or sufficiently large), the optimal $a_j$ will be exactly or close to 0 or 1. Consequently, regularization and constraints offer interpretability and flexibility, which allows us to use more accurate models to promote structures in the neural networks, and the proposed convergent ProxSGD algorithm ensures efficient training of such models.

## 2 THE PROPOSED PROXSGD ALGORITHM

In this section, we describe the ProxSGD algorithm to solve (1).

*Background and setup.* We make the following blanket assumptions on problem (1).

- $f_i(\boldsymbol{x}, \boldsymbol{y}^{(i)})$ is smooth (continuously differentiable) but not necessarily convex.
- $\nabla_{\boldsymbol{x}} f_i(\boldsymbol{x}, \boldsymbol{y}^{(i)})$ is Lipschitz continuous with a finite constant $L_i$ for any $\boldsymbol{y}_i$. Thus $\nabla f(\boldsymbol{x})$ is Lipschitz continuous with constant $L \triangleq \frac{1}{m}\sum_{i=1}^{m} L_i$.
- $r(\boldsymbol{x})$ is convex, proper and lower semicontinuous (not necessarily smooth).
- $\mathbb{X}$ is convex and compact.

We are interested in algorithms that can find a stationary point of (1). A stationary point $\boldsymbol{x}^\star$ satisfies the optimality condition: at $\boldsymbol{x} = \boldsymbol{x}^\star$,

$$(\boldsymbol{x} - \boldsymbol{x}^\star)^T \nabla f(\boldsymbol{x}^\star) + r(\boldsymbol{x}) - r(\boldsymbol{x}^\star) \ge 0, \forall \boldsymbol{x} \in \mathbb{X}. \tag{4}$$

When $r(\boldsymbol{x}) = 0$ and $\mathbb{X} = \mathbb{R}^n$, the deterministic optimization problem 1 can be solved by the (batch) gradient descent method. When $m$, the number of training examples, is large, it is computationally expensive to calculate the gradient. Instead, we estimate the gradient by a minibatch of $m(t)$ training examples. We denote the minibatch by $\mathbb{M}(t)$: its elements are drawn uniformly from $\{1, 2, \ldots, m\}$ and there are $m(t)$ elements. Then the estimated gradient is

$$\boldsymbol{g}(t) \triangleq \frac{1}{m(t)}\sum_{i\in\mathbb{M}(t)} \nabla f_i(\boldsymbol{x}(t), \boldsymbol{y}^{(i)}) \tag{5}$$

and it is an unbiased estimate of the true gradient.

*The proposed algorithm.* The instantaneous gradient $\boldsymbol{g}(t)$ is used to form an aggregate gradient (momentum) $\boldsymbol{v}(t)$, which is updated recursively as follows

$$\boldsymbol{v}(t) = (1 - \rho(t))\boldsymbol{v}(t-1) + \rho(t)\boldsymbol{g}(t), \tag{6}$$

where $\rho(t)$ is the stepsize (learning rate) for the momentum and $\rho(t) \in (0, 1]$.

At iteration $t$, we propose to solve an approximation subproblem and denote its solution as $\widehat{\boldsymbol{x}}(\boldsymbol{x}(t), \boldsymbol{v}(t), \boldsymbol{\tau}(t))$, or simply $\widehat{\boldsymbol{x}}(t)$

$$\widehat{\boldsymbol{x}}(t) \triangleq \underset{\boldsymbol{x}\in\mathbb{X}}{\arg\min} \left\{ (\boldsymbol{x} - \boldsymbol{x}(t))^T \boldsymbol{v}(t) + \frac{1}{2}(\boldsymbol{x} - \boldsymbol{x}(t))^T \text{diag}(\boldsymbol{\tau}(t))(\boldsymbol{x} - \boldsymbol{x}(t)) + r(\boldsymbol{x}) \right\}. \tag{7}$$

A quadratic regularization term is incorporated so that the subproblem (7) is strongly convex and its modulus is the minimum element of the vector $\boldsymbol{\tau}(t)$, denoted as $\underline{\tau}(t)$ and $\underline{\tau}(t) = \min_{j=1,\ldots,n} \tau_j(t)$. Note that $\underline{\tau}(t)$ should be lower bounded by a positive constant that is strictly larger than 0, so that the quadratic regularization in (7) will not vanish.

The difference between two vectors $\widehat{\boldsymbol{x}}(t)$ and $\boldsymbol{x}(t)$ specifies a direction starting at $\boldsymbol{x}(t)$ and ending at $\widehat{\boldsymbol{x}}(t)$. This update direction is used to refine the weight vector

$$\boldsymbol{x}(t+1) = \boldsymbol{x}(t) + \epsilon(t)(\widehat{\boldsymbol{x}}(t) - \boldsymbol{x}(t)), \tag{8}$$

| algorithm | momentum | weight | quadratic gain in subproblem | regularization | constraint set |
|---|---|---|---|---|---|
| ProxSGD | $\rho(t)$ | $\epsilon(t)$ | $\boldsymbol{\tau}(t)$ | convex | convex, compact |
| SGD (w. momentum) | $1(\rho)$ | $\epsilon$ | $\mathbf{1}$ | 0 | $\mathbb{R}^n$ |
| AdaGrad | 1 | $\epsilon$ | $\sqrt{\mathbf{r}(t)} + \delta\mathbf{1}^{\dagger}$ | 0 | $\mathbb{R}^n$ |
| RMSProp | 1 | $\epsilon$ | $\sqrt{\mathbf{r}(t)} + \delta\mathbf{1}^{\ddagger}$ | 0 | $\mathbb{R}^n$ |
| ADAM | $\rho$ | $\frac{\epsilon}{1-\rho^t}$ | $\sqrt{\frac{\mathbf{r}(t)}{1-\beta^t}} + \delta\mathbf{1}^{\ddagger}$ | 0 | $\mathbb{R}^n$ |
| AMSGrad | $\rho$ | $\epsilon$ | $\sqrt{\widehat{\boldsymbol{r}}(t)}$ 
 $\widehat{\boldsymbol{r}}(t) = \max(\widehat{\boldsymbol{r}}(t-1), \boldsymbol{r}(t))^{\ddagger}$ | 0 | $\mathbb{R}^n$ |
| ADABound | $\rho$ | 1 | $\frac{1}{\mathrm{clip}(\epsilon(t)/\sqrt{\boldsymbol{r}(t)}, \boldsymbol{\eta}_l(t), \boldsymbol{\eta}_u(t))}^{\ddagger}$ | 0 | $\mathbb{R}^n$ |

Table 1: Connection between the proposed framework and existing methods, where $\rho, \beta, \epsilon$ and $\delta$ are some predefined constants. $^{\dagger}\mathbf{r}(t) = \mathbf{r}(t-1) + \mathbf{g}(t) \odot \mathbf{g}(t)$, $^{\ddagger}\mathbf{r}(t) = \beta\mathbf{r}(t-1) + (1-\beta)\mathbf{g}(t) \odot \mathbf{g}(t)$.

where $\epsilon(t)$ is a stepsize (learning rate) for the weight and $\epsilon(t) \in (0, 1]$. Note that $\boldsymbol{x}(t+1)$ is feasible as long as $\boldsymbol{x}(t)$ is feasible, as it is the convex combination of two feasible points $\boldsymbol{x}(t)$ and $\widehat{\boldsymbol{x}}(t)$ while the set $\mathbb{X}$ is convex.

The above steps (5)-(8) are summarized in Algorithm 1, which is termed proximal-type Stochastic Gradient Descent (ProxSGD), for the reason that the explicit constraint $x \in \mathbb{X}$ in (7) can also be formulated implicitly as a regularization function, more specifically, the indicator function $\delta_{\mathbb{X}}(\boldsymbol{x})$. If all elements of $\boldsymbol{\tau}(t)$ are equal, then $\widehat{\boldsymbol{x}}(t)$ is exactly the proximal operator

$$\widehat{\boldsymbol{x}}(t) = \underset{\boldsymbol{x}}{\mathrm{argmin}} \left\{ \left\| \boldsymbol{x} - \left( \boldsymbol{x}(t) - \frac{1}{\tau(t)}\boldsymbol{v}(t) \right) \right\|_2^2 + \frac{1}{\tau(t)}r(\boldsymbol{x}) + \delta_{\mathbb{X}}(\boldsymbol{x}) \right\}$$

$$\triangleq \mathrm{Prox}_{\frac{1}{\tau(t)}r(\boldsymbol{x})+\delta_{\mathbb{X}}(\boldsymbol{x})} \left( \boldsymbol{x}(t) - \frac{1}{\tau(t)}\boldsymbol{v}(t) \right).$$

---

**Algorithm 1** Proximal-type Stochastic Gradient Descent (ProxSGD) Method

---

**Input:** $\boldsymbol{x}(0) \in \mathbb{X}$, $\boldsymbol{v}(-1) = \mathbf{0}$, $t = 0, T$, $\{\rho(t)\}_{t=0}(t)$, $\{\epsilon(t)\}_{t=0}(t)$.
**for** $t = 0 : 1 : T$ **do**

1. Compute the instantaneous gradient $\boldsymbol{g}(t)$ based on the minibatch $\mathbb{M}(t)$:

$$\boldsymbol{g}(t) = \frac{1}{m(t)} \sum_{i \in \mathbb{M}(t)} \nabla_{\boldsymbol{x}} f_i(\boldsymbol{x}(t), \boldsymbol{y}^{(i)}).$$

2. Update the momentum: $\boldsymbol{v}(t) = (1 - \rho(t))\boldsymbol{v}(t-1) + \rho(t)\boldsymbol{g}(t)$.

3. Compute $\widehat{\boldsymbol{x}}(t)$ by solving the approximation subproblem:

$$\widehat{\boldsymbol{x}}(t) = \underset{\boldsymbol{x} \in \mathbb{X}}{\arg\min} \left\{ (\boldsymbol{x} - \boldsymbol{x}(t))(t)\boldsymbol{v}(t) + \frac{1}{2}(\boldsymbol{x} - \boldsymbol{x}(t))^T \mathrm{diag}(\boldsymbol{\tau}(t))(\boldsymbol{x} - \boldsymbol{x}(t)) + r(\boldsymbol{x}) \right\}.$$

4. Update the weight: $\boldsymbol{x}(t+1) = \boldsymbol{x}(t) + \epsilon(t)(\widehat{\boldsymbol{x}}(t) - \boldsymbol{x}(t))$.

**end for**

---

ProxSGD in Algorithm 1 bears a similar structure as several SGD algorithms, without and with momentum, see Table 1, and it allows to interpret some existing algorithms as special cases of the proposed framework. For example, no momentum is used in SGD, and this amounts to setting $\rho(t) = 1$ in Algorithm 1. In ADAM, the learning rate for momentum is a constant $\rho$ and the learning rate for the weight vector is given by $\epsilon/(1 - \rho^t)$ for some $\epsilon$, and this simply amounts to setting $\rho(t) = \rho$ and $\epsilon(t) = \epsilon/(1-\rho^t)$ in Algorithm 1. This interpretation also implies that the convergence conditions to be proposed shortly later are also suffcient for existing algorithms (although they are not meant to be the weakest conditions available in literature).

*Solving the approximation subproblem (7).* Since (7) is strongly convex, $\widehat{\boldsymbol{x}}(t)$ is unique. Generally $\widehat{\boldsymbol{x}}(t)$ in (7) does not admit a closed-form expression and should be solved by a generic solver. However, some important special cases that are frequently used in practice can be solved efficiently.

• The trivial case is $\mathbb{X} = \mathbb{R}^n$ and $r = 0$, where

$$\widehat{\boldsymbol{x}}(t) = \boldsymbol{x}(t) - \frac{\boldsymbol{v}(t)}{\boldsymbol{\tau}(t)}, \tag{9}$$

where the vector division is understood to be element-wise. When $\mathbb{X} = \mathbb{R}^n$ and $r(\boldsymbol{x}) = \mu\|\boldsymbol{x}\|_1$, $\widehat{\boldsymbol{x}}(t)$ has a closed-form expression that is known as the soft-thresholding operator

$$\widehat{\boldsymbol{x}}(t) = S_{\frac{\mu\mathbf{1}}{\boldsymbol{\tau}(t)}} \left( \boldsymbol{x}(t) - \frac{\boldsymbol{v}(t)}{\boldsymbol{\tau}(t)} \right), \tag{10}$$

where $S_{\boldsymbol{a}}(\boldsymbol{b}) \triangleq \max(\boldsymbol{b} - \boldsymbol{a}, \boldsymbol{0}) - \max(-\boldsymbol{b} - \boldsymbol{a}, \boldsymbol{0})$ (Bach et al., 2011).

• If $\mathbb{X} = \mathbb{R}^n$ and $r(\boldsymbol{x}) = \mu\|\boldsymbol{x}\|_2$ and $\boldsymbol{\tau}(t) = \tau\boldsymbol{I}$ for some $\tau$, then (Parikh & Boyd, 2014)

$$\widehat{\boldsymbol{x}}(t) = \begin{cases} (1 - \mu/\|\tau\boldsymbol{x}(t) - \boldsymbol{v}(t)\|_2)(\boldsymbol{x}(t) - \boldsymbol{v}(t)/\tau), & \text{if } \|\tau\boldsymbol{x}(t) - \boldsymbol{v}(t)\|_2 \geq \mu, \\ 0, & \text{otherwise.} \end{cases} \tag{11}$$

If $\boldsymbol{x}$ is divided into blocks $\boldsymbol{x}_1, \boldsymbol{x}_2, \ldots$, the $\ell_2$-regularization is commonly used to promote block sparsity (rather than element sparsity by $\ell_1$-regularization).

• When there is a bound constraint $\boldsymbol{l} \leq \boldsymbol{x} \leq \boldsymbol{u}$, $\widehat{\boldsymbol{x}}(t)$ can simply be obtained by first solving the approximation subproblem (7) without the bound constraint and then projecting the optimal point onto the interval $[\boldsymbol{l}, \boldsymbol{u}]$. For example, when $\mathbb{X} = \mathbb{R}^n$ and $r = 0$,

$$\widehat{\boldsymbol{x}}(t) = \left[ \boldsymbol{x}(t) - \frac{\boldsymbol{v}(t)}{\boldsymbol{\tau}(t)} \right]_{\boldsymbol{l}}^{\boldsymbol{u}}, \tag{12}$$

with $[\boldsymbol{x}]_{\boldsymbol{l}}^{\boldsymbol{u}} = \text{clip}(\boldsymbol{x}, \boldsymbol{l}, \boldsymbol{u}) \triangleq \min(\max(\boldsymbol{x}, \boldsymbol{l}), \boldsymbol{u})$.

• If the constraint function is quadratic: $\mathbb{X} = \{\boldsymbol{x} : \|\boldsymbol{x}\|_2^2 \leq 1\}$, $\widehat{\boldsymbol{x}}(t)$ has a semi-analytical expression (up to a scalar Lagrange multiplier which can be found efficiently by the bisection method).

*Approximation subproblem.* We explain why we update the weights by solving an approximation subproblem (7). First, we denote $\widetilde{f}$ as the smooth part of the objective function in (7). Clearly it depends on $\boldsymbol{x}(t)$ and $\boldsymbol{v}(t)$ (and thus $\mathbb{M}(t)$), while $\boldsymbol{x}(t)$ and $\boldsymbol{v}(t)$ depend on the old weights $\boldsymbol{x}(0), \ldots, \boldsymbol{x}(t-1)$ and momentum and $\boldsymbol{v}(0), \ldots, \boldsymbol{v}(t-1)$. Define $\mathbb{F}(t) \triangleq \{\boldsymbol{x}(0), \ldots, \boldsymbol{x}(t), \mathbb{M}(0), \ldots, \mathbb{M}(t)\}$ as a shorthand notation for the trajectory generated by ProxSGD. We formally write $\widetilde{f}$ as

$$\widetilde{f}(\boldsymbol{x}; \mathbb{F}(t)) \triangleq (\boldsymbol{x} - \boldsymbol{x}(t))^T\boldsymbol{v}(t) + \frac{1}{2}(\boldsymbol{x} - \boldsymbol{x}(t))^T\text{diag}(\boldsymbol{\tau}(t))(\boldsymbol{x} - \boldsymbol{x}(t)). \tag{13}$$

It follows from the optimality of $\widehat{\boldsymbol{x}}(t)$ that

$$\widetilde{f}(\boldsymbol{x}(t); \mathbb{F}(t)) + r(\boldsymbol{x}(t)) \geq \widetilde{f}(\widehat{\boldsymbol{x}}(t); \mathbb{F}(t)) + r(\widehat{\boldsymbol{x}}(t)).$$

After inserting (13) and reorganizing the terms, the above inequality becomes

$$(\widehat{\boldsymbol{x}}(t) - \boldsymbol{x}(t))^T\boldsymbol{v}(t) + r(\widehat{\boldsymbol{x}}(t)) - r(\boldsymbol{x}(t)) \leq -\tau(t)\|\widehat{\boldsymbol{x}}(t) - \boldsymbol{x}(t)\|_2^2. \tag{14}$$

Since $\nabla f(\boldsymbol{x})$ is Lipschitz continuous with constant $L$, we have

$$f(\boldsymbol{x}(t+1)) + r(\boldsymbol{x}(t+1)) - (f(\boldsymbol{x}(t)) + r(\boldsymbol{x}(t)))$$

$$\leq (\boldsymbol{x}(t+1) - \boldsymbol{x}(t))^T\nabla f(\boldsymbol{x}(t)) + \frac{L}{2}\|\boldsymbol{x}(t+1) - \boldsymbol{x}(t)\|_2^2 + r(\boldsymbol{x}(t+1)) - r(\boldsymbol{x}(t)) \tag{15}$$

$$\leq \epsilon(t)\left( (\widehat{\boldsymbol{x}}(t) - \boldsymbol{x}(t))^T\nabla f(\boldsymbol{x}(t)) + r(\widehat{\boldsymbol{x}}(t)) - r(\boldsymbol{x}(t)) + \frac{L}{2}\epsilon(t)\|\widehat{\boldsymbol{x}}(t) - \boldsymbol{x}(t)\|_2^2 \right), \tag{16}$$

where the first inequality follows from the descent lemma (applied to $f$) and the second inequality follows from the Jensen's inequality of the convex function $r$ and the update rule (8).

If $\boldsymbol{v}(t) = \nabla f(\boldsymbol{x}(t))$ (which is true asymptotically as we show shortly later), by replacing $\nabla f(\boldsymbol{x}(t))$ in (16) by $\boldsymbol{v}(t)$ and inserting (14) into (16), we obtain

$$f(\boldsymbol{x}(t+1)) + r(\boldsymbol{x}(t+1)) - (f(\boldsymbol{x}(t)) + r(\boldsymbol{x}(t))) \leq \epsilon(t) \left( \frac{L}{2}\epsilon(t) - \tau(t) \right) \|\widehat{\boldsymbol{x}}(t) - \boldsymbol{x}(t)\|_2^2. \quad (17)$$

The right hand side (RHS) will be negative when $\epsilon(t) < \frac{2\tau(t)}{L}$: this will eventually be satisfied as we shall use a decaying $\epsilon(t)$. This implies that the proposed update (8) will decrease the objective value of (1) after each iteration.

*Momentum and algorithm convergence.* It turns out that the momentum (gradient averaging step) in (6) is essential for the convergence of ProxSGD. Under some mild technical assumptions we outline now, the aggregate gradient $\boldsymbol{v}(t)$ will converge to the true (unknown) gradient $\nabla f(\boldsymbol{x}(t))$. This remark is made rigorous in the following theorem.

**Theorem 1.** *Assume that the unbiased gradient $\boldsymbol{g}(t)$ has a bounded second moment*

$$\mathbb{E}\left[\|\boldsymbol{g}(t)\|_2^2 \,|\mathbb{F}(t)\right] \leq C, \quad (18)$$

*for some finite and positive constant C, and the sequence of stepsizes $\{\rho(t)\}$ and $\{\epsilon(t)\}$ satisfy*

$$\sum_{t=0}^{\infty} \rho(t) = \infty, \sum_{t=0}^{\infty} \rho(t)^2 < \infty, \sum_{t=0}^{\infty} \epsilon(t) = \infty, \sum_{t=0}^{\infty} \epsilon(t)^2 < \infty, \lim_{t\to\infty} \frac{\epsilon(t)}{\rho(t)} = 0. \quad (19)$$

*Then $\lim_{t\to\infty} \|\boldsymbol{v}(t) - \nabla f(\boldsymbol{x}(t))\| = 0$, and every limit point of the sequence $\{\boldsymbol{x}(t)\}$ is a stationary point of (1) w.p.1.*

*Proof.* Under the assumptions (18) and (19), it follows from Lemma 1 of Ruszczyński (1980) that $\boldsymbol{v}(t) \to \nabla f(\boldsymbol{x}(t))$. Since the descent direction $\widehat{\boldsymbol{x}}(t) - \boldsymbol{x}(t)$ is a descent direction in view of (14), the convergence of the ProxSGD algorithm can be obtained by generalizing the line of analysis in Theorem 1 of Yang et al. (2016) for smooth optimization problems. The detailed proof is included in the appendix to make the paper self-contained. □

We draw some comments on the convergence analysis in Theorem 1.

The bounded second moment assumption on the gradient $\boldsymbol{g}$ in (18) and decreasing stepsizes in (19) are standard assumptions in stochastic optimization and SGD. What is noteworthy is that $\epsilon(t)$ should decrease faster than $\rho(t)$ to ensure that $\boldsymbol{v}(t) \to \nabla f(\boldsymbol{x}(t))$. But this is more of an interest from the theoretical perspective, and in practice, we observe that $\epsilon(t)/\rho(t) = a$ for some constant $a$ that is smaller than 1 usually yields satisfactory performance, as we show numerically in the next section.

According to Theorem 1, the momentum $\boldsymbol{v}(t)$ converges to the (unknown) true gradient $\nabla f(\boldsymbol{x}(t))$, so the ProxSGD algorithm eventually behaves similar to the (deterministic) gradient descent algorithm. This property is essential to guarantee the convergence of the ProxSGD algorithm.

To guarantee the theoretical convergence, the quadratic gain $\underline{\tau}(t)$ in the approximation subproblem (7) should be lower bounded by some positive constant (and it does not even have to be time-varying). In practice, there are various rationales to define it (see Table 1), and they lead to different empirical convergence speed and generalization performance.

The technical assumptions in Theorem 1 may not always be fully satisfied by the neural networks deployed in practice, due to, e.g., the nonsmooth ReLU activation function, batch normalization and dropout. Nevertheless, Theorem 1 still provides valuable guidance on the algorithm's practical performance and the choice of the hyperparameters.

## 3 SIMULATION RESULTS

In this section, we perform numerical experiments to test the proposed ProxSGD algorithm[1]. In particular, we first train two SNN to compare ProxSGD with ADAM (Kingma & Ba, 2015), AMS-Grad (Reddi et al., 2018), ADABound (Luo et al., 2019) and SGD with momentum. Then we train

---

[1]The simulations in Setion 3.1 and 3.3 are implemented in TensorFlow and available at `https://github.com/optyang/proxsgd`. The simulations in Section 3.2 are implemented in PyTorch and available at `https://github.com/cc-hpc-itwm/proxsgd`.

a BNN to illustrate the merit of regularization and constraints. To ensure a fair comparison, the hyperparameters of all algorithms are chosen according to either the inventors' recommendations or a hyperparameter search. Furthermore, in all simulations, the quadratic gain $\boldsymbol{\tau}(t)$ in ProxSGD is updated in the same way as ADAM, with $\beta = 0.999$ (see Table 1).

### 3.1 SPARSE NEURAL NETWORK: TRAINING CONVOLUTION NEURAL NETWORKS ON CIFAR-10

We first consider the multiclass classification problem on CIFAR-10 dataset (Krizhevsky, 2009) with convolution neural network (CNN). The network has 6 convolutional layers and each of them is followed by a batch normalization layer; the exact setting is shown in Table 2.

Table 2: CNN Settings

| parameter | value |
|---|---|
| data set | CIFAR-10 |
| number of convolution layers | 6 |
| size of convolution kernels | 3×3 |
| number of output filters in convolution layers 1-2, 3-4, 5-6 | 32, 64, 128 |
| operations after convolution layers 1-2, 3-4, 5-6 | max pooling, dropout (rate=0.2) |
| kernel size, stride, padding of maxing pooling | 2×2, 2, valid |
| activation function for convolution/output layer | elu/softmax |
| loss function and regularization function | cross entropy and $\ell_1$-norm |

Following the parameter configurations of ADAM in Kingma & Ba (2015), AMSGrad in Reddi et al. (2018), and ADABound in Luo et al. (2019), we set $\rho = 0.1$, $\beta = 0.999$ and $\epsilon = 0.001$ (see Table 1), which are uniform for all the algorithms and commonly used in practice. Note that we have also activated $\ell_1$-regularization for these algorithms in the built-in function in TensorFlow/PyTorch, which amounts to adding the subgradient of the $\ell_1$-norm to the gradient of the loss function. For the proposed ProxSGD, $\epsilon(t)$ and $\rho(t)$ decrease over the iterations as follows,

$$\epsilon(t) = \frac{0.06}{(t+4)^{0.5}}, \ \ \rho(t) = \frac{0.9}{(t+4)^{0.5}}. \tag{20}$$

Recall that the $\ell_1$-norm in the approximation subproblem naturally leads to the soft-thresholding proximal mapping, see (10). The regularization parameter $\mu$ in the soft-thresholding then permits controlling the sparsity of the parameter variable $\boldsymbol{x}$; in this experiment we set $\mu = 5 \cdot 10^{-5}$.

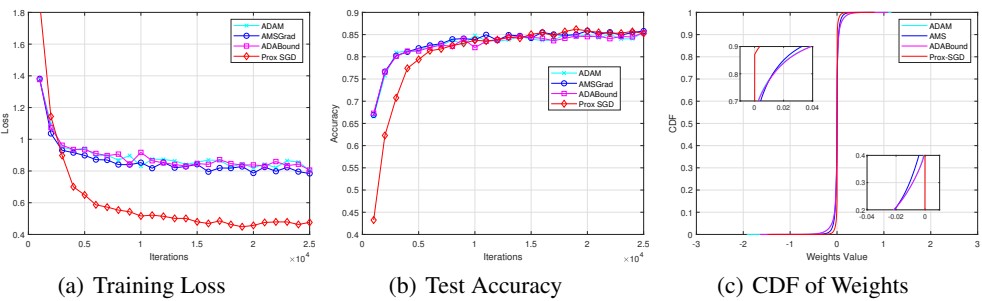

(a) Training Loss      (b) Test Accuracy      (c) CDF of Weights

Figure 1: Performance comparison for CNN on CIFAR-10.

In Figure 1, we compare the four algorithms (ProxSGD, ADAM, AMSGrad, ADABound) in terms of three metrics, namely, the training loss, the test accuracy and the achieved sparsity. On the one hand, Figure 1(a) shows that ProxSGD outperforms ADAM, AMSGrad and ADABound in the achieved loss value. On the other hand, the achieved accuracy is comparable, see Figure 1(b).

The sparsity of the trained model is measured by the cumulative distribution function (CDF) of the weights' value, which specifies the percentage of weights below any given value. For the proposed

ProxSGD in Figure 1(c), we can observe at 0 in the x-axis the abrupt change of the CDF in the y-axis, which implies that more than 90% of the weights are *exactly* zero. By comparison, only 40%-50% are exactly zero by the other algorithms. What is more, for this experiment, the soft-thresholding proximal operator in ProxSGD does not increase the computation time: ADAM 17.24s (per epoch), AMSGrad 17.44s, ADABound 16.38s, ProxSGD 16.04s. Therefore, in this experiment, the proposed ProxSGD with soft-thresholding proximal mapping has a clear and substantial advantage than other stochastic subgradient-based algorithms.

### 3.2 SPARSE NEURAL NETWORK: TRAINING DENSENET-201 ON CIFAR-100

In this subsection, the performance of ProxSGD is evaluated by a much more complex network and dataset. In particular, we train the DenseNet-201 network (Huang et al., 2017) for CIFAR-100 (Krizhevsky, 2009). DenseNet-201 is the deepest topology of the DenseNet family and belongs to the state of the art networks in image classification tasks. We train the network using Prox-SGD, ADAM and SGD with momentum. To ensure a fair comparison among these algorithms, the learning rate is not explicitly decayed during training for all algorithms. Furthermore, the ideal hyperparameters for each algorithm were computed by grid-search and the curves are averaged over five runs. A batch-size of 128 is adopted. For ProxSGD, the regularization parameter is $\mu = 10^{-5}$, the learning rate for the weight and momentum is, respectively,

$$\epsilon(t) = \frac{0.15}{(t+4)^{0.5}}, \ \ \rho(t) = \frac{0.9}{(t+4)^{0.5}}.$$

For ADAM, $\epsilon = 6 \cdot 10^{-4}$ and $\rho = 0.1$. SGD with momentum uses a learning rate of $\epsilon = 6 \cdot 10^{-3}$ and a momentum of 0.9 (so $\rho = 0.1$). The regularization parameter for both ADAM and SGD with momentum is $\mu = 10^{-4}$.

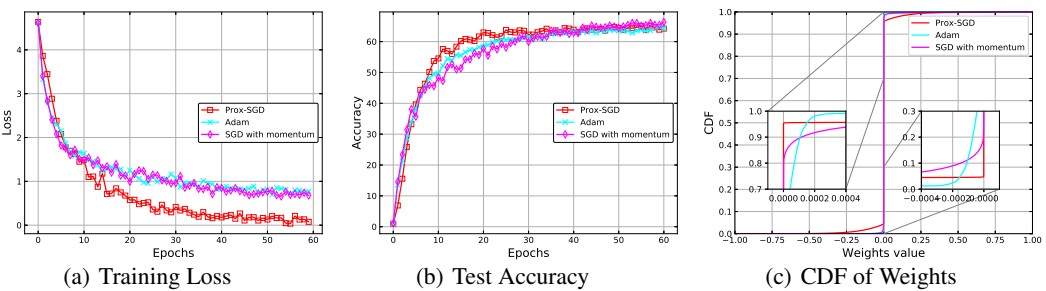

(a) Training Loss      (b) Test Accuracy      (c) CDF of Weights

Figure 2: Performance comparison for DenseNet-201 on CIFAR-100.

Figure 2 shows the performance of ProxSGD and other algorithms for DenseNet-201 trained on CIFAR-100. We see from Figure 2(a) that ProxSGD achieves the lowest training loss after Epoch 10. The test accuracy in Figure 2(b) shows that all algorithms achieve similar accuracy and ProxSGD outperforms the other two during the early stage of training. We remark that this is achieved with a much sparser network as shown in Figure 2(c). In particular, we can see from the zoomed-in part of Figure 2(c) that SGD with momentum has approximately 70% of their weights at zero, while most weights learned by ADAM are not exactly zero (although they are very small). In contrast, ProxSGD reaches the sparsity of 92-94%.

In Figure 3, we demonstrate that ProxSGD is much more efficient in generating a SNN, irrespective of the hyperparameters (related to the learning rate). In particular, we try many different initial learning rate of the weight vector $\epsilon(0)$ for ProxSGD and test their performance. From Figure 3(a)-(b) we see that, as expected, the hyperparameters affect the achieved training loss and test accuracy, and many lead to a worse training loss and/or test accuracy than ADAM and SGD with momentum. However, Figure 3(c) shows that most of them (except when they are too small: $\epsilon(0) = 0.01$ and 0.001) generate a much sparser NN than both ADAM and SGD with momentum. These observations are also consistent with the theoretical framework in Section 2: interpretating ADAM and SGD with

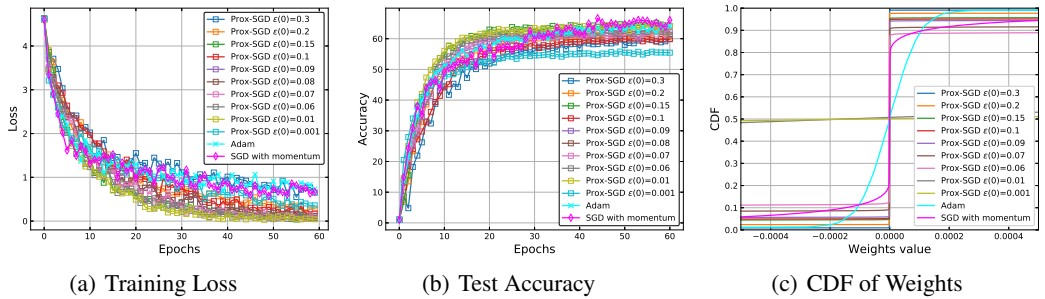

(a) Training Loss      (b) Test Accuracy      (c) CDF of Weights

Figure 3: Hyperparameters and sparsity for DenseNet-201 on CIFAR-100.

momentum as special cases of ProxSGD implies that they have the same convergence rate, and the sparsity is due to the explicit use of the nonsmooth $\ell_1$-norm regularization.

For this experiment, the soft-thresholding proximal operator in ProxSGD increases the training time: the average time per epoch for ProxSGD is 3.5 min, SGD with momentum 2.8 min and ADAM 2.9 min. In view of the higher level of sparsity achieved by ProxSGD, this increase in computation time is reasonable and affordable.

## 3.3 BINARY NEURAL NETWORKS: TRAINING DEEP NEURAL NETWORKS ON MNIST

In this subsection, we evaluate the proposed algorithm ProxSGD in training the BNN by solving problem (3). We train a 6-layer fully-connected deep neural network (DNN) for the MNIST dataset, and we use the tanh activation function to promote a binary activation output; see Table 3. The algorithm parameters are the same as Sec. 3.1, except that $\mu = 2 \cdot 10^{-4}$. The chosen setup is particularly suited to evaluate the merit of the proposed method, since MNIST is a simple dataset and it allows us to investigate soly the effect of the proposed model and training algorithm.

Table 3: DNN Settings

| parameter | Value |
|---|---|
| dataset | MNIST |
| number of hidden layers | 6 |
| number of nodes per hidden layer | 200 |
| activation function in hidden/output layer | tanh/softmax |
| loss function | cross entropy |

After customizing Algorithm 1 to problem (3), the approximation subproblem is

$$(\widehat{\boldsymbol{x}}(t), \widehat{\boldsymbol{a}}(t)) = \underset{\boldsymbol{0} \leq \boldsymbol{a} \leq \boldsymbol{1}}{\arg\min} \left\{ \begin{array}{l} (\boldsymbol{x} - \boldsymbol{x}(t))^T \boldsymbol{v}_x(t) + \frac{1}{2}(\boldsymbol{x} - \boldsymbol{x}(t))^T \mathrm{diag}(\boldsymbol{\tau}_x(t))(\boldsymbol{x} - \boldsymbol{x}(t)) \\ + (\boldsymbol{a} - \boldsymbol{a}(t))^T \boldsymbol{v}_a(t) + \frac{1}{2}(\boldsymbol{a} - \boldsymbol{a}(t))^T \mathrm{diag}(\boldsymbol{\tau}_a(t))(\boldsymbol{a} - \boldsymbol{a}(t)) \end{array} \right\}.$$

Both $\widehat{\boldsymbol{x}}(t)$ and $\widehat{\boldsymbol{a}}(t)$ have a closed-form expression (cf. (9) and (12))

$$\widehat{\boldsymbol{x}}(t) = \boldsymbol{x}(t) - \frac{\boldsymbol{v}_x(t)}{\boldsymbol{\tau}_x(t)}, \text{ and } \widehat{\boldsymbol{a}}(t) = \left[ \boldsymbol{a}(t) - \frac{\boldsymbol{v}_a(t)}{\boldsymbol{\tau}_a(t)} \right]_{\boldsymbol{0}}^{\boldsymbol{1}}, \tag{21}$$

where $\boldsymbol{v}_x(t)$ and $\boldsymbol{v}_a(t)$ are the momentum updated in the spirit of (6), with the gradients given by

$$\boldsymbol{g}_x(t) = \frac{1}{m(t)} \sum_{i \in \mathbb{M}(t)} \nabla f_i(\boldsymbol{x}(t), \boldsymbol{y}^{(i)}) + \frac{\mu}{2}(\boldsymbol{x}(t) + 2\boldsymbol{a}(t) - \boldsymbol{1}), \text{ and } \boldsymbol{g}_a(t) = \mu \boldsymbol{x}(t).$$

The training loss is shown in Figure 4(a). We remark that during the training process of ProxSGD, the weights are not binarized, for the reason that the penalty should regularize the problem in a way such that the optimal weights (to which ProxSGD converges) are exactly or close to 1 or -1. After training is completed, the CDF of the learned weights is summarized in Figure 4(c), and then the

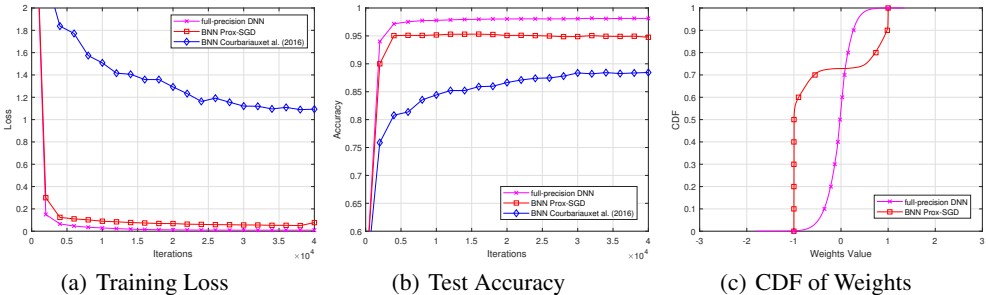

| (a) Training Loss | (b) Test Accuracy | (c) CDF of Weights |

Figure 4: Performance comparison for BNN on MNIST

learned weights are binarized to generate a full BNN whose test accuracy is in Figure 4(b). On the one hand, we see from Figure 4(a)-(b) that the achieved training loss and test accuracy by BNN is worse than the standard full-precision DNN (possibly with soft-thresholding). This is expected as BNN imposes regularization and constraints on the optimization problem and reduces the search space. However, the difference in test accuracy is quite small. On the other hand, we see from Figure 4(c) that the regularization in the proposed formulation (3) is very effective in promoting binary weights: 15% of weights are in the range (-1,-0.5) and 15% of weights are in the range (0.5,1), and all the other weights are either -1 or 1. As all weights are exactly or close to 1 or -1, we could just binarize the weights to exactly 1 or -1 only once by hard thresholding, after the training is completed, and thus the incurred performance loss is small (98% versus 95% for test accuracy). In contrast, the weights generated by the full-precision DNN (that is, without regularization) are smoothly distributed in $[-2, 2]$.

Even though the proposed formulation (3) doubles the number of parameters to optimize (from $\boldsymbol{x}$ in full-precision DNN to $(\boldsymbol{x}, \boldsymbol{a})$ in BNN ProxSGD), the convergence speed is equally fast in terms of the number of iterations. The computation time is also roughly the same: full-precision DNN 13.06s (per epoch) and ProxSGD 12.21s. We remark that $\boldsymbol{g}_a(t)$, the batch gradient w.r.t. $\boldsymbol{a}$, has a closed-form expression and it does not involve the back-propagation. In comparison with the algorithm in Courbariaux et al. (2016), the proposed ProxSGD converges much faster and achieves a much better training loss and test accuracy (95% versus 89%, the computation time per epoch for Courbariaux et al. (2016) is 13.56s). The notable performance improvement is due to the regularization and constraints. Naturally we should make an effort of searching for a proper regularization parameter $\mu$, but this effort is very well paid off. Furthermore, we observe in the simulations that the performance is not sensitive to the exact value of $\mu$, as long as it is in an appropriate range.

## 4 CONCLUDING REMARKS

In this paper, we proposed ProxSGD, a proximal-type stochastic gradient descent algorithm with momentum, for constrained optimization problems where the smooth loss function is augmented by a nonsmooth and convex regularization. We considered two applications, namely the stochastic training of SNN and BNN, to show that regularization and constraints can effectively promote structures in the learned network. More generally, incorporating regularization and constraints allows us to use a more accurate and interpretable model for the problem at hand and the proposed convergent ProxSGD algorithms ensures efficient training. Numerical tests showed that ProxSGD outperforms state-of-the-art algorithms, in terms of convergence speed, achieved training loss and/or the desired structure in the learned neural networks.

### ACKNOWLEDGEMENT

The work of Yang is supported by DFG Project DeTol. The work of van Sloun is part of the research programme Rubicon ENW 2018-3 with project number 019.183EN.014, which is financed by the Dutch Research Council (NWO). The work of Yuan, Lei and Chatzinotas is supported by ERC AGNOSTIC, FNR CORE ASWELL and FNR-AFR LARGOS.

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

## A Appendix: Proof of Theorem 1

*Proof.* The claim $\lim_{t \to \infty} \|\boldsymbol{v}(t) - \nabla f(\boldsymbol{x}(t))\| = 0$ is a consequence of (Ruszczyński, 1980, Lemma 1). To see this, we just need to verify that all the technical conditions therein are satisfied by the problem at hand. Specifically, Condition (a) of (Ruszczyński, 1980, Lemma 1) is satisfied because $\mathbb{X}$ is closed and bounded. Condition (b) of (Ruszczyński, 1980, Lemma 1) is exactly (18). Conditions (c)-(d) of (Ruszczyński, 1980, Lemma 1) come from the stepsize rules in (19) of Theorem 1. Condition (e) of (Ruszczyński, 1980, Lemma 1) comes from the Lipschitz property of $\nabla f$ and stepsize rule in (19) of Theorem 1.

We need the following intermediate result to prove the limit point of the sequence $\boldsymbol{x}(t)$ is a stationary point of (1).

**Lemma 1.** *There exists a constant $\widehat{L}$ such that*

$$\|\widehat{\boldsymbol{x}}(\boldsymbol{x}(t_1), \boldsymbol{\xi}(t_1)) - \widehat{\boldsymbol{x}}(\boldsymbol{x}(t_2), \boldsymbol{\xi}(t_2))\| \leq \widehat{L}\|\boldsymbol{x}(t_1) - \boldsymbol{x}(t_2)\| + e(t_1, t_2),$$

*and $\lim_{t_1, t_2 \to \infty} e(t_1, t_2) = 0$ w.p.1.*

*Proof.* We assume without loss of generality (w.l.o.g.) that $\boldsymbol{\tau}(t) = \tau\mathbf{1}$, and the approximation subproblem (7) reduces to

$$\widehat{\boldsymbol{x}}(t) \triangleq \underset{\boldsymbol{x} \in \mathbb{X}}{\operatorname{argmin}} \left\{ (\boldsymbol{x} - \boldsymbol{x}(t))^T \boldsymbol{v}(t) + \frac{\tau}{2}\|\boldsymbol{x} - \boldsymbol{x}(t)\|_2^2 + r(\boldsymbol{x}) \right\}.$$

It is further equivalent to

$$\min_{\boldsymbol{x} \in \mathbb{X}, r(\boldsymbol{x}) \leq y} \left\{ (\boldsymbol{x} - \boldsymbol{x}(t))^T \boldsymbol{v}(t) + \frac{\tau}{2}\|\boldsymbol{x} - \boldsymbol{x}(t)\|_2^2 + y \right\}, \tag{22}$$

where the (unique) optimal $\boldsymbol{x}$ and $y$ is $(\widehat{\boldsymbol{x}}(t)$ and $r(\widehat{\boldsymbol{x}}(t))$, respectively.

We assume w.l.o.g. that $t_2 > t_1$. It follows from first-order optimality condition that

$$(\boldsymbol{x} - \widehat{\boldsymbol{x}}(t_1))^T (\boldsymbol{v}(t_1) + \tau(\widehat{\boldsymbol{x}}(t_1) - \boldsymbol{x}(t_1))) + y - r(\widehat{\boldsymbol{x}}(t_1)) \geq 0, \forall \boldsymbol{x}, y \text{ such that } r(\boldsymbol{x}) \leq y \tag{23a}$$

$$(\boldsymbol{x} - \widehat{\boldsymbol{x}}(t_2))^T (\boldsymbol{v}(t_2) + \tau(\widehat{\boldsymbol{x}}(t_2) - \boldsymbol{x}(t_2))) + y - r(\widehat{\boldsymbol{x}}(t_2)) \geq 0, \forall \boldsymbol{x}, y \text{ such that } r(\boldsymbol{x}) \leq y. \tag{23b}$$

Setting $(\boldsymbol{x}, y) = (\widehat{\boldsymbol{x}}(t_2), r(\widehat{\boldsymbol{x}}(t_2)))$ in (23a) and $(\boldsymbol{x}, y) = (\widehat{\boldsymbol{x}}(t_1), r(\widehat{\boldsymbol{x}}(t_1)))$ in (23b), and adding them up, we obtain

$$(\widehat{\boldsymbol{x}}(t_1) - \widehat{\boldsymbol{x}}(t_2))^T (\boldsymbol{v}(t_1) - \boldsymbol{v}(t_2)) - \tau(\boldsymbol{x}(t_1) - \boldsymbol{x}(t_2))^T (\widehat{\boldsymbol{x}}(t_1) - \widehat{\boldsymbol{x}}(t_2)) \leq -\tau\|\widehat{\boldsymbol{x}}(t_1) - \widehat{\boldsymbol{x}}(t_2)\|_2^2. \tag{24}$$

The term on the left hand side can be lower bounded as follows:

$$\langle \widehat{\boldsymbol{x}}(t_1) - \widehat{\boldsymbol{x}}(t_2), \boldsymbol{v}(t_1) - \nabla f(\boldsymbol{x}(t_1)) - \boldsymbol{v}(t_2) + \nabla f(\boldsymbol{x}(t_2)) \rangle$$
$$+ \langle \widehat{\boldsymbol{x}}(t_1) - \widehat{\boldsymbol{x}}(t_2), \nabla f(\boldsymbol{x}(t_1)) - \nabla f(\boldsymbol{x}(t_2)) \rangle - \tau\langle \widehat{\boldsymbol{x}}(t_1) - \widehat{\boldsymbol{x}}(t_2), \boldsymbol{x}(t_1) - \boldsymbol{x}(t_2) \rangle$$
$$\geq -\|\widehat{\boldsymbol{x}}(t_1) - \widehat{\boldsymbol{x}}(t_2)\|(\varepsilon(t_1) + \varepsilon(t_2)) - (L + \tau)\|\widehat{\boldsymbol{x}}(t_1) - \widehat{\boldsymbol{x}}(t_2)\|\|\boldsymbol{x}(t_1) - \boldsymbol{x}(t_2)\| \tag{25}$$

where the inequality comes from the Lipschitz continuity of $\nabla f(\boldsymbol{x})$, with $\varepsilon(t) \triangleq \|\boldsymbol{v}(t) - \nabla f(\boldsymbol{x}(t))\|$.

Combining the inequalities (24) and (25), we have

$$\|\widehat{\boldsymbol{x}}(t_1) - \widehat{\boldsymbol{x}}(t_2)\| \leq (L + \tau)\tau^{-1}\|\boldsymbol{x}(t_1) - \boldsymbol{x}(t_2)\| + \tau^{-1}(\varepsilon(t_1) + \varepsilon(t_2)),$$

which leads to the desired (asymptotic) Lipschitz property:

$$\|\widehat{\boldsymbol{x}}(t_1) - \widehat{\boldsymbol{x}}(t_2)\| \leq \widehat{L}\|\boldsymbol{x}(t_1) - \boldsymbol{x}(t_2)\| + e(t_1, t_2),$$

with $\widehat{L} \triangleq \tau^{-1}(L + \tau)$ and $e(t_1, t_2) \triangleq \tau^{-1}(\varepsilon(t_1) + \varepsilon(t_2))$, and $\lim_{t_1 \to \infty, t_2 \to \infty} e(t_1, t_2) = 0$ w.p.1. $\qquad \square$

Define $U(\boldsymbol{x}) \triangleq f(\boldsymbol{x}) + r(\boldsymbol{x})$. Following the line of analysis from (15) to (16), we obtain

$$U(\boldsymbol{x}(t+1)) - U(\boldsymbol{x}(t)) \tag{26}$$

$$\leq \epsilon(t)((\widehat{\boldsymbol{x}}(t) - \boldsymbol{x}(t))^T(\nabla f(\boldsymbol{x}(t)) + r(\widehat{\boldsymbol{x}}(t)) - r(\boldsymbol{x}(t))) + \frac{L}{2}\epsilon(t)^2\|\widehat{\boldsymbol{x}}(t) - \boldsymbol{x}(t)\|^2$$

$$= \epsilon(t)(\widehat{\boldsymbol{x}}(t) - \boldsymbol{x}(t))^T(\nabla f(\boldsymbol{x}(t)) - \boldsymbol{v}(t) + \boldsymbol{v}(t) + r(\widehat{\boldsymbol{x}}(t)) - r(\boldsymbol{x}(t))) + \frac{L}{2}\epsilon(t)^2\|\widehat{\boldsymbol{x}}(t) - \boldsymbol{x}(t)\|^2$$

$$\leq -\epsilon(t)\left(\tau - \frac{L}{2}\epsilon(t)\right)\|\widehat{\boldsymbol{x}}(t) - \boldsymbol{x}(t)\|^2 + \epsilon(t)\|\widehat{\boldsymbol{x}}(t) - \boldsymbol{x}(t)\|\|\nabla f(\boldsymbol{x}(t)) - \boldsymbol{v}(t)\|, \tag{27}$$

where in the last inequality we used (14) and the Cauchy-Schwarz inequality.

Let us show by contradiction that $\liminf_{t\to\infty}\|\widehat{\boldsymbol{x}}(t) - \boldsymbol{x}(t)\| = 0$ w.p.1. Suppose $\liminf_{t\to\infty}\|\widehat{\boldsymbol{x}}(t) - \boldsymbol{x}(t)\| \geq \chi > 0$ with a positive probability. Then we can find a realization such that at the same time $\|\widehat{\boldsymbol{x}}(t) - \boldsymbol{x}(t)\| \geq \chi > 0$ for all $t$ and $\lim_{t\to\infty}\|\nabla f(\boldsymbol{x}(t)) - \boldsymbol{v}(t)\| = 0$; we focus next on such a realization. Using $\|\widehat{\boldsymbol{x}}(t) - \boldsymbol{x}(t)\| \geq \chi > 0$, the inequality (27) is equivalent to

$$U(\boldsymbol{x}(t+1)) - U(\boldsymbol{x}(t)) \leq -\epsilon(t)\left(\tau - \frac{L}{2}\epsilon(t) - \frac{1}{\chi}\|\nabla f(\boldsymbol{x}(t)) - \boldsymbol{v}(t)\|\right)\|\widehat{\boldsymbol{x}}(t) - \boldsymbol{x}(t)\|^2. \tag{28}$$

Since $\lim_{t\to\infty}\|\nabla f(\boldsymbol{x}(t)) - \boldsymbol{v}(t)\| = 0$, there exists a $t_0$ sufficiently large such that

$$\tau - \frac{L}{2}\epsilon(t) - \frac{1}{\chi}\|\nabla f(\boldsymbol{x}(t)) - \boldsymbol{v}(t)\| \geq \bar{\tau} > 0, \quad \forall t \geq t_0. \tag{29}$$

Therefore, it follows from (28) and (29) that

$$U(\boldsymbol{x}(t)) - U(\boldsymbol{x}^{t_0}) \leq -\bar{\tau}\chi^2 \sum_{n=t_0}(t)\epsilon^{n+1}, \tag{30}$$

which, in view of $\sum_{n=t_0}^{\infty} \epsilon^{n+1} = \infty$, contradicts the boundedness of $\{U(\boldsymbol{x}(t))\}$. Therefore it must be $\liminf_{t\to\infty}\|\widehat{\boldsymbol{x}}(t) - \boldsymbol{x}(t)\| = 0$ w.p.1.

Let us show by contradiction that $\limsup_{t\to\infty}\|\widehat{\boldsymbol{x}}(t) - \boldsymbol{x}(t)\| = 0$ w.p.1. Suppose $\limsup_{t\to\infty}\|\widehat{\boldsymbol{x}}(t) - \boldsymbol{x}(t)\| > 0$ with a positive probability. We focus next on a realization along with $\limsup_{t\to\infty}\|\widehat{\boldsymbol{x}}(t) - \boldsymbol{x}(t)\| > 0$, $\lim_{t\to\infty}\|\nabla f(\boldsymbol{x}(t)) - \boldsymbol{v}(t)\| = 0$, $\liminf_{t\to\infty}\|\widehat{\boldsymbol{x}}(t) - \boldsymbol{x}(t)\| = 0$, and $\lim_{t_i,t_2\to\infty} e(t_1, t_2) = 0$, where $e(t_1, t_2)$ is defined in Lemma 1. It follows from $\limsup_{t\to\infty}\|\widehat{\boldsymbol{x}}(t) - \boldsymbol{x}(t)\| > 0$ and $\liminf_{t\to\infty}\|\widehat{\boldsymbol{x}}(t) - \boldsymbol{x}(t)\| = 0$ that there exists a $\delta > 0$ such that $\|\triangle\boldsymbol{x}(t)\| \geq 2\delta$ (with $\triangle\boldsymbol{x}(t) \triangleq \widehat{\boldsymbol{x}}(t) - \boldsymbol{x}(t)$) for infinitely many $t$ and also $\|\triangle\boldsymbol{x}(t)\| < \delta$ for infinitely many $t$. Therefore, one can always find an infinite set of indexes, say $\mathcal{T}$, having the following properties: for any $t \in \mathcal{T}$, there exists an integer $i_t > t$ such that

$$\|\triangle\boldsymbol{x}(t)\| < \delta, \quad \|\triangle\boldsymbol{x}(i_t)\| > 2\delta, \quad \delta \leq \|\triangle\boldsymbol{x}(n)\| \leq 2\delta, t < n < i_t. \tag{31}$$

Given the above bounds, the following holds: for all $t \in \mathcal{T}$,

$$\begin{aligned}
\delta &\leq \|\triangle\boldsymbol{x}(i_t)\| - \|\triangle\boldsymbol{x}(t)\| \\
&\leq \|\triangle\boldsymbol{x}(i_t) - \triangle\boldsymbol{x}(t)\| = \|(\widehat{\boldsymbol{x}}(i_t) - \boldsymbol{x}(i_t)) - (\widehat{\boldsymbol{x}}(t) - \boldsymbol{x}(t))\| \\
&\leq \|\widehat{\boldsymbol{x}}(i_t) - \widehat{\boldsymbol{x}}(t)\| + \|\boldsymbol{x}(i_t) - \boldsymbol{x}(t)\| \\
&\leq (1+\widehat{L})\|\boldsymbol{x}(i_t) - \boldsymbol{x}(t)\| + e(i_t, t) \\
&\leq (1+\widehat{L})\sum_{n=t}^{i_t-1}\epsilon(n)\|\triangle\boldsymbol{x}(n)\| + e(i_t, t) \\
&\leq 2\delta(1+\widehat{L})\sum_{n=t}^{i_t-1}\epsilon(n) + e(i_t, t),
\end{aligned} \tag{32}$$

implying that

$$\liminf_{\mathcal{T}\ni t\to\infty} \sum_{n=t}^{i_t-1}\epsilon(n) \geq \bar{\delta}_1 \triangleq \frac{1}{2(1+\widehat{L})} > 0. \tag{33}$$

Proceeding as in (32), we also have: for all $t \in \mathcal{T}$,

$$\|\triangle\boldsymbol{x}(t+1)\| - \|\triangle\boldsymbol{x}(t)\| \leq \|\triangle\boldsymbol{x}(t+1) - \triangle\boldsymbol{x}(t)\| \leq (1+\widehat{L})\epsilon(t)\|\triangle\boldsymbol{x}(t)\| + e(t, t+1),$$

which leads to

$$(1 + (1 + \widehat{L})\epsilon(t)) \|\triangle \boldsymbol{x}(t)\| + e(t, t+1) \geq \|\triangle \boldsymbol{x}(t+1)\| \geq \delta, \tag{34}$$

where the second inequality follows from (31). It follows from (34) that there exists a $\bar{\delta}_2 > 0$ such that for sufficiently large $t \in \mathcal{T}$,

$$\|\triangle \boldsymbol{x}(t)\| \geq \frac{\delta - e(t, t+1)}{1 + (1 + \widehat{L})\epsilon(t)} \geq \bar{\delta}_2 > 0. \tag{35}$$

Here after we assume w.l.o.g. that (35) holds for all $t \in \mathcal{T}$ (in fact one can always restrict $\{\boldsymbol{x}(t)\}_{t \in \mathcal{T}}$ to a proper subsequence).

We show now that (33) is in contradiction with the convergence of $\{U(\boldsymbol{x}(t))\}$. Invoking (27), we have for all $t \in \mathcal{T}$,

$$
\begin{aligned}
U(\boldsymbol{x}(t+1)) - U(\boldsymbol{x}(t)) &\leq -\epsilon(t)\left(\tau - \frac{L}{2}\epsilon(t)\right)\left\|\widehat{\boldsymbol{x}}(t) - \boldsymbol{x}(t)\right\|^2 + \epsilon(t)\delta\|\nabla f(\boldsymbol{x}(t)) - \boldsymbol{v}(t)\| \\
&\leq -\epsilon(t)\left(\tau - \frac{L}{2}\epsilon(t) - \frac{\|\nabla f(\boldsymbol{x}(t)) - \boldsymbol{v}(t)\|}{\delta}\right)\left\|\widehat{\boldsymbol{x}}(t) - \boldsymbol{x}(t)\right\|^2 \\
&\quad + \epsilon(t)\delta\|\nabla f(\boldsymbol{x}(t)) - \boldsymbol{v}(t)\|^2,
\end{aligned} \tag{36}
$$

and for $t < n < i_t$,

$$
\begin{aligned}
U(\boldsymbol{x}(n+1)) - U(\boldsymbol{x}(n)) &\leq -\epsilon(n)\left(\tau - \frac{L}{2}\epsilon(n) - \frac{\|\nabla f(\boldsymbol{x}(n)) - \boldsymbol{v}(n)\|}{\|\widehat{\boldsymbol{x}}(n) - \boldsymbol{x}(n)\|}\right)\left\|\widehat{\boldsymbol{x}}(n) - \boldsymbol{x}(n)\right\|^2 \\
&\leq -\epsilon(n)\left(\tau - \frac{L}{2}\epsilon(n) - \frac{\|\nabla f(\boldsymbol{x}(n)) - \boldsymbol{v}(n)\|}{\delta}\right)\left\|\widehat{\boldsymbol{x}}(n) - \boldsymbol{x}(n)\right\|^2,
\end{aligned} \tag{37}
$$

where the last inequality follows from (31). Adding (36) and (37) over $n = t+1, \ldots, i_t - 1$ and, for $t \in \mathcal{T}$ sufficiently large (so that $\tau - L\epsilon(t)/2 - \delta^{-1}\|\nabla f(\boldsymbol{x}(n)) - \boldsymbol{v}(n)\| \geq \widehat{\tau} > 0$ and $\|\nabla f(\boldsymbol{x}(t)) - \boldsymbol{v}(t)\| < \widehat{\tau}\bar{\delta}_2^2/\delta$), we have

$$
\begin{aligned}
U(\boldsymbol{x}(i_t)) - U(\boldsymbol{x}(t)) &\overset{(a)}{\leq} -\widehat{\tau}\sum_{n=t}^{i_t-1}\epsilon(n)\left\|\widehat{\boldsymbol{x}}(n) - \boldsymbol{x}(n)\right\|^2 + \epsilon(t)\delta\|\nabla f(\boldsymbol{x}(t)) - \boldsymbol{v}(t)\| \\
&\overset{(b)}{\leq} -\widehat{\tau}\bar{\delta}_2^2\sum_{n=t+1}^{i_t-1}\epsilon(n) - \epsilon(t)\left(\widehat{\tau}\bar{\delta}_2^2 - \delta\|\nabla f(\boldsymbol{x}(t)) - \boldsymbol{v}(t)\|\right) \\
&\overset{(c)}{\leq} -\widehat{\tau}\bar{\delta}_2^2\sum_{n=t+1}^{i_t-1}\epsilon(n),
\end{aligned} \tag{38}
$$

where (a) follows from $\tau - L\epsilon(t)/2 - \delta^{-1}\|\nabla f(\boldsymbol{x}(n)) - \boldsymbol{v}(n)\| \geq \widehat{\tau} > 0$; (b) is due to (35); and in (c) we used $\|\nabla f(\boldsymbol{x}(t)) - \boldsymbol{v}(t)\| < \widehat{\tau}\bar{\delta}_2^2/\delta$. Since $\{U(\boldsymbol{x}(t))\}$ converges, it must be $\liminf_{\mathcal{T} \ni t \to \infty} \sum_{n=t+1}^{i_t-1}\epsilon(n) = 0$, which contradicts (33). Therefore, it must be $\limsup_{t \to \infty}\|\widehat{\boldsymbol{x}}(t) - \boldsymbol{x}(t)\| = 0$ w.p.1.

Finally, let us prove that every limit point of the sequence $\{\boldsymbol{x}(t)\}$ is a stationary solution of (1). Let $\boldsymbol{x}^\star$ be the limit point of the convergent subsequence $\{\boldsymbol{x}(t)\}_{t \in \mathcal{T}}$. Taking the limit of (23a) over the index set $\mathcal{T}$ (and replacing w.l.o.g. $y$ by $r(\boldsymbol{x})$), we have

$$
\begin{aligned}
&\lim_{\mathcal{T} \ni t \to \infty} (\boldsymbol{x} - \widehat{\boldsymbol{x}}(t))^T (\boldsymbol{v}(t) + \tau(\widehat{\boldsymbol{x}}(t) - \boldsymbol{x}(t))) + r(\boldsymbol{x}) - r(\widehat{\boldsymbol{x}}(t)) \\
&= (\boldsymbol{x} - \boldsymbol{x}^\star)^T \nabla f(\boldsymbol{x}^\star) + r(\boldsymbol{x}) - r(\boldsymbol{x}^\star) \geq 0, \ \forall \boldsymbol{x} \in \mathbb{X},
\end{aligned}
$$

where the last equality follows from: i) $\lim_{t \to \infty}\|\nabla f(\boldsymbol{x}(t)) - \boldsymbol{v}(t)\| = 0$, and ii) $\lim_{t \to \infty}\|\widehat{\boldsymbol{x}}(t) - \boldsymbol{x}(t)\| = 0$. This is the desired first-order optimality condition and $\boldsymbol{x}^\star$ is a stationary point of (1). $\square$

