# OpenReview forum: "ProxSGD: Training Structured Neural Networks under Regularization and Constraints"
_ICLR.cc/2020/Conference — Accept (Poster)_

### Official Review · AnonReviewer3 · 2019-10-23
**Official Blind Review #3**

**Rating:** 3

**Review:**


[Summary]
This paper proposes Prox-SGD, a theoretical framework for stochastic optimization algorithms that (1) incorporates momentum and coordinate-wise scaling as in Adam, and (2) can handle constraint and (non-smooth) regularizers through the proximal operator. With proper choices of hyperparameters, the algorithm is shown to converge asymptotically to stationarity, for smooth non-convex loss + convex constraint/regularizer. The algorithm is empirically tested on training binary and sparse neural nets on MNIST and CIFAR-10.

[Pros]
The theoretical framework that incorporates most of the commonly used tweaks in stochastic optimization for deep learning, and a convergence result that establishes broadly the asymptotic convergence to stationarity.

[Cons]
The result of Theorem 1 sounds rather a straightforward application of classical results; important sub-cases such as Adam violates the assumption (Adam has \rho_t = \rho so \sum \rho_t^2 = \infty) and thus are not contained in this case. From this it seems to me Theorem 1 says things mostly about the “easier” sub-cases, and thus is perhaps not very surprising and a bit limited in bringing in new messages.

The experiments are mostly done on simple problems --- 3 of the 4 figures are on MNIST. The specific tasks (training sparse / binary neural nets with MLP / vanilla CNN architectures) considered in the experiments are all very extensively studied in prior work, and the results in this paper says at most that the proposed Prox-SGD works for these tasks.

Overall, I like the idea in this paper that we can put together a unified framework for stochastic optimization algorithms and incorporate things like momentums and regularizations that were previously treated separately. However, beyond proposing such a framework, it seems that contributions on both the theoretical and empirical side are a bit limited at this point.

***

Thank the authors for the response and the efforts in revising the paper. I am glad to see the additional experiments for training sparse networks on CIFAR-100 (a much harder task than MNIST and also CIFAR-10) in which the proposed method works well. This largely resolved my concerns on the experimental side.

However, I'd still like to hold my evaluation on the theoretical side, in that approaches for handling constraints / non-smoothness / momentums are fairly well understood in the optimization literature. The present result (Theorem 1) conveys the message that these approaches can be combined to work (give an algorithm that converges to stationary points if it converges), but is not really a result that gives us new understandings / novel proof techniques beyond that.

I have slightly improved my rating to reflect my updated evaluation.


**Experience Assessment:**

I have published one or two papers in this area.

**Review Assessment: Checking Correctness Of Derivations And Theory:**

I assessed the sensibility of the derivations and theory.

**Review Assessment: Checking Correctness Of Experiments:**

I assessed the sensibility of the experiments.

**Review Assessment: Thoroughness In Paper Reading:**

I read the paper at least twice and used my best judgement in assessing the paper.

---

> ### Author Response · Authors · 2019-11-15
> **We thank the reviewer for taking time to perform a thorough and constructive review.**
>
>
> Comment: The result of Theorem 1 sounds rather a straightforward application of classical results; important sub-cases such as Adam violates the assumption (Adam has $\rho_t = \rho$ so $\sum \rho_t^2 = \infty$) and thus are not contained in this case. From this it seems to me Theorem 1 says things mostly about the “easier” sub-cases, and thus is perhaps not very surprising and a bit limited in bringing in new messages.
>
> Reply: We respectfully disagree with the reviewer.
>
> The intention of Theorem 1 (on the convergence of the proposed Prox-SGD algorithm) is not to propose new convergence conditions that are weaker than state-of-the-art conditions such as ADAM for unconstrained and smooth optimization problems. On the contrary, it is to provide a convergence guarantee for the proposed Prox-SGD algorithm which can be applied to train neural networks under nonsmooth regularization and constraints. Despite the fact that the proof largely follows from classical results, it does not compromise the importance of the theorem which provides the convergence analysis for the proposed Prox-SGD algorithm.
>
> As a matter of fact, nonsmooth regularization and constraints are difficult subcases that have not been addressed before. To our best knowledge, the proposed Prox-SGD is the first convergent algorithm proposed to train neural networks under nonsmooth regularization and constraints. We have revised the paper accordingly to make our motivations and contributions more clear.
>
> ***
>
> Comment: The experiments are mostly done on simple problems --- 3 of the 4 figures are on MNIST. The specific tasks (training sparse / binary neural nets with MLP / vanilla CNN architectures) considered in the experiments are all very extensively studied in prior work, and the results in this paper says at most that the proposed Prox-SGD works for these tasks.
>
> Reply: We thank the reviewer for the constructive comment.
>
> In this paper, we aim at proposing a stochastic algorithm that is applicable for a wide class of problems, rather than proposing application-specific solutions. We chose sparse and binary neural networks as example applications as they are important and challenging. The performance of the proposed Prox-SGD algorithm is illustrated by the excellent empirical results and the potential is thus consolidated. Consequently, our work contains significant contributions to both theory and applications.
>
> For sparse neural networks, motivated by the reviewer’s comment, we removed the MNIST experiments and included a new experiment on DenseNet-201 for CIFAR-100. Compared to the benchmark algorithms, the proposed Prox-SGD achieves a much sparse neural network without any loss in the training error and test accuracy.
>
> For the binary neural networks, although MNIST is a simple dataset, state-of-the-art algorithm does not deliver satisfactory performance. Therefore this experiment still delivers an important message: the notable performance improvement demonstrates the advantage of incorporating constraints into the neural networks, which can only be trained efficiently by the proposed algorithm.
>
> ***
>
> Comment: Overall, I like the idea in this paper that we can put together a unified framework for stochastic optimization algorithms and incorporate things like momentums and regularizations that were previously treated separately. However, beyond proposing such a framework, it seems that contributions on both the theoretical and empirical side are a bit limited at this point.
>
> Reply: We thank the reviewer for the comment.
>
> We would like to stress again that we intend to propose a stochastic algorithm that can train neural networks with nonsmooth regularization and constraints. Momentum is essential to guarantee the convergence, and this is in sharp contrast to unconstrained optimization problems where momentum is only optional.
>
> From the theoretical perspective, we have proposed the first convergent stochastic algorithm for training neural networks with nonsmooth regularization and constraints.
>
> To strengthen our empirical contribution, we included a new experiment on DenseNet201 for CIFAR-100. For the example applications considered in this paper, the proposed Prox-SGD can (1) generate a neural network that is much sparser than state-of-the-art algorithms, and (2) generate a binary neural net that has a much higher accuracy than state-of-the-art algorithms.
>
> ***
>
> We thank the reviewer again for the valuable comments. We have revised the paper accordingly to make the above points more clear. We kindly and respectfully ask the reviewer to consider updating the rating if the comments are addressed to the reviewer’s satisfaction.

---

### Official Review · AnonReviewer1 · 2019-10-29
**Official Blind Review #1**

**Rating:** 6

**Review:**

The paper proposes a new gradient-based stochastic optimization algorithm (with gradient averaging) by adapting theory for proximal algorithms (originally developed for convex problems) to the non-convex setting. The main idea is to first use an averaged gradient plus a quadratic term to locally approximate the non-convex function with a convex smooth function before applying the proximal operator on it. As a result, the algorithm will be able to solve non-smooth (e.g., l1-regularized) and constrained non-convex problems, which will be very useful for optimization problems arising from deep learning.

I think this is a potentially good paper that proposes an algorithm for wide applicability. But I still see some issues that prompts me to ask the following questions:

1. What is the time complexity of solving the sub convex problem at every iteration? The authors did not discuss this in the experiments, but this is very important in evaluating the applicability of the proposed algorithm especially on large-scale problems.
2. The authors should provide more explanation on the term \tau (t): It seems it is only used as an auxiliary parameter in the quadratic approximation of f, and that it doesn't affect the convergence asymptotically. But I would imagine it would affect the practical convergence at the beginning of the algorithm?
3. The authors have repeatedly mentioned that using the averaged gradient v(t) is very important for the convergence analysis of the algorithm. But I did not see how this is the case from the analysis discussed in the paper (I didn't check the proof in reference Ruszczynski (1980)). As this is important in justifying the algorithm, I think the authors should include a discussion to provide some intuition on this in the paper.
4. Line (15): should vx(t) be x(t) instead? If not, where does the term come from?


**Experience Assessment:**

I have published one or two papers in this area.

**Review Assessment: Checking Correctness Of Derivations And Theory:**

I assessed the sensibility of the derivations and theory.

**Review Assessment: Checking Correctness Of Experiments:**

I assessed the sensibility of the experiments.

**Review Assessment: Thoroughness In Paper Reading:**

I read the paper at least twice and used my best judgement in assessing the paper.

---

> ### Author Response · Authors · 2019-11-15
> **We sincerely appreciate the reviewer for performing a thorough and constructive review. We also thank the reviewer’s positive feedback to our contributions.**
>
>
> 1. What is the time complexity of solving the sub convex problem at every iteration? The authors did not discuss this in the experiments, but this is very important in evaluating the applicability of the proposed algorithm especially on large-scale problems.
>
> Reply: We thank the reviewer for the constructive comment.
>
> Our algorithm needs to perform the soft-thresholding operation, but the other algorithms need to compute the subgradient of L1 norm function. To address this comment, we have measured the epoch time for the experiment on CNN and CIFAR-10 in Sec. 3.2. The computation time per epoch is: ADAM 17.24s, AMSGrad 17.44s, ADABound 16.38s, Prox-SGD 16.04s.
>
> For the experiment on DenseNet-201 and CIFAR-100, the soft-thresholding operation in Prox-SGD increases the training time: the average time per epoch for Prox-SGD is 3.5min, SGD with momentum 2.8min and ADAM 2.9min. In view of the higher level of sparsity achieved by Prox-SGD (92-94%) compared with SGD with momentum (70%) and ADAM (20-30%), this increase in computation is reasonable and affordable.
>
> For the experiment on BNN, the proposed Prox-SGD has closed-form updates. Furthermore, the gradient with respect to a has a closed-form expression as well and it does not involve back-propogation. The average time per epoch for full-precision DNN and Prox-SGD is roughly the same: 13.06s and 12.21s. So doubling the problem parameters in Prox-SGD does not seem to increase the computational complexity.
> These observations are also included in the revised paper.
>
> ***
>
> 2. The authors should provide more explanation on the term $\tau(t)$: It seems it is only used as an auxiliary parameter in the quadratic approximation of $f$, and that it doesn't affect the convergence asymptotically. But I would imagine it would affect the practical convergence at the beginning of the algorithm?
>
> Reply: We thank the reviewer for pointing out this issue.
>
> On the one hand, to guarantee the theoretical convergence, it is sufficient that it is lower bounded by a constant that is larger than 0. It is not even necessarily time varying. On the other hand, we agree with the reviewer that the choice of tau would impact the algorithm’s empirical performance. In our paper, we did not propose a new method to update tau. In our simulations, we adopted the same rule to update tau as ADAM.
>
> We have revised the discussion after the theorem to make the above point clear and emphasized again in the simulations.
>
> ***
>
> 3. The authors have repeatedly mentioned that using the averaged gradient $v(t)$ is very important for the convergence analysis of the algorithm. But I did not see how this is the case from the analysis discussed in the paper (I didn't check the proof in reference Ruszczynski (1980)). As this is important in justifying the algorithm, I think the authors should include a discussion to provide some intuition on this in the paper.
>
> Reply: We thank the reviewer for the constructive comment. The averaged gradient $v(t)$ will converge to the true (but unknown) gradient of the loss function $f$, and the stochastic algorithm will eventually resembles the (deterministic) gradient descent algorithm. This property is very important in establishing the convergence.
>
> Although an intuitive explanation was given in (13)-(17), we have revised the paper to make the above point more clear. In particular, we added a comment after Theorem 1 to discuss its importance. We also included the proof to make the paper self-contained.
>
> ***
>
> 4. Line (15): should $vx(t)$ be $x(t)$ instead? If not, where does the term come from?
>
> Reply: We thank the reviewer for pointing out this typo. We have corrected this typo and proofread the whole paper again.
>
> ***
>
> We thank the reviewer again for the constructive comments. We kindly and respectfully ask the reviewer to consider updating the rating if the comments are addressed to the reviewer’s satisfaction.

---

### Official Review · AnonReviewer2 · 2019-10-30
**Official Blind Review #2**

**Rating:** 6

**Review:**

Summary:

This paper introduces and analyzes a training algorithm for neural networks with non smooth regularization and weight constraints (such as sparsity or binarization). The analysis shows that, under assumptions, the proposed algorithm converges almost surely to a stationary point. Experimental results show that the proposed algorithm can train both sparse and binary neural networks.

Major comments:

This paper presents an interesting combination of theoretical analysis and experiments demonstrating the benefits of the proposed training algorithm. Because the setting considered is fairly general, it is likely to be widely useful in a variety of settings that have up to now been approached with case-specific algorithms (eg training binary NNs).

It should be noted that objective functions for many NNs are not smooth (eg ReLU-based networks, which are used in the experiments) and the convergence argument does not apply directly to these.

The paper could be improved by more extensively optimizing hyper parameters of all algorithms in the experimental evaluation. From the current experiments, it is not possible to evaluate whether performance differences are coming from differences in learning rate schedules, etc, or from the proposed algorithm specifically. For instance, the competitor algorithms are run with fixed momentum and learning rate parameters, while the proposed algorithm is run with a handcrafted decay in these parameters. Ideally, the hyperparameters should be optimized out such that each algorithm uses its best settings. At minimum, the choice of hyperparameter schedule for the algorithm should be justified.

The paper could also be strengthened by comparing the runtime of the proposed algorithm to prior methods. Prox SGD trains faster in terms of iterations (hyper parameter differences aside), but how about wall clock time? This is particularly important in the binary case where additional optimization parameters are added and updated in each iteration.

The main theoretical result is presented with a sketch of a proof, and I did not attempt to reconstruct the argument from the named sources. It could be useful to provide a full proof (perhaps in an appendix) to allow the work to be self-contained.

The paper is clearly written and easy to follow.


Typos:
In general more care should be taken with the equations:
Eq. 7 x^t should be x(t)
Eq. 7 differs slightly from Alg. 1 step 3 because of the \mu term in step 3. I would remove \mu
Pg 3, the indicator function is introduced as \sigma_X then put in the equation as \delta_X

**Experience Assessment:**

I have published one or two papers in this area.

**Review Assessment: Checking Correctness Of Derivations And Theory:**

I assessed the sensibility of the derivations and theory.

**Review Assessment: Checking Correctness Of Experiments:**

I carefully checked the experiments.

**Review Assessment: Thoroughness In Paper Reading:**

I read the paper at least twice and used my best judgement in assessing the paper.

---

> ### Author Response · Authors · 2019-11-15
> **We sincerely appreciate the reviewer for performing a thorough and constructive review. We also thank the reviewer’s positive feedback to our contributions.**
>
>
> Comment: It should be noted that objective functions for many NNs are not smooth (eg ReLU-based networks, which are used in the experiments) and the convergence argument does not apply directly to these.
>
> Reply: The smoothness of the loss function is a common assumption in convergence analysis, and there is some work showing empirically the loss function tends to be smooth even when the nonsmooth ReLu activation function is used (see Figure 1 and the discussion in https://arxiv.org/pdf/1712.09913 for example). But we agree with the reviewer that the theoretical assumptions may not always be fully satisfied by practical NNs due to, e.g., ReLU, batch normalization and dropout. We have included such a remark after Theorem 1 in the revision.
>
> ***
>
> Comment: The paper could be improved by more extensively optimizing hyper parameters of all algorithms in the experimental evaluation. From the current experiments, it is not possible to evaluate whether performance differences are coming from differences in learning rate schedules, etc, or from the proposed algorithm specifically. For instance, the competitor algorithms are run with fixed momentum and learning rate parameters, while the proposed algorithm is run with a handcrafted decay in these parameters. Ideally, the hyperparameters should be optimized out such that each algorithm uses its best settings. At minimum, the choice of hyperparameter schedule for the algorithm should be justified.
>
> Reply: We agree with the reviewer that all algorithms’s convergence speed and achieved accuracy depend heavily on the choice of hyperparameters.
>
> On the one hand, to ensure a fair comparison, the hyperparameters of the benchmark algorithms are chosen either according to the inventors’ recommendations, or according to a hyperparameter grid search. We have included such a remark in the revision. Besides, please note that Prox-SGD can converge to a very sparse solution and this accelerates the convergence to a certain extent.
>
> On the other hand, we pay special attention to the special structure promoted by the proposed Prox-SGD algorithm, namely, nonsmooth regularization and constraints. The achieved empirical improvement is predicted and supported by theory. To show this, we include a new simulation on DenseNet-201 and CIFAR-100 (Figure 3 in the revised paper) which shows that the proposed Prox-SGD with different hyperparameters all generate a very sparse NN, although they lead to different training loss and test accurary.
>
> ***
>
> Comment: The paper could also be strengthened by comparing the runtime of the proposed algorithm to prior methods. Prox SGD trains faster in terms of iterations (hyper parameter differences aside), but how about wall clock time? This is particularly important in the binary case where additional optimization parameters are added and updated in each iteration.
>
> Reply: We thank the reviewer for the constructive comment.
>
> Our algorithm needs to perform the soft-thresholding operation, but the other algorithms need to compute the subgradient of L1 norm function. To address this comment, we have measured the epoch time for the experiment on CNN and CIFAR-10 in Sec. 3.2. The computation time per epoch is: ADAM 17.24s, AMSGrad 17.44s, ADABound 16.38s, Prox-SGD 16.04s.
>
> For the experiment on DenseNet-201 and CIFAR-100, the soft-thresholding operation in Prox-SGD increases the training time: the average time per epoch for Prox-SGD is 3.5min, SGD with momentum 2.8min and ADAM 2.9min. In view of the higher level of sparsity achieved by Prox-SGD (92-94%) compared with SGD with momentum (70%) and ADAM (20-30%), this increase in computation is reasonable and affordable.
>
> For the experiment on BNN, the proposed Prox-SGD has closed-form updates. Furthermore, the gradient with respect to the vector variable $a$ has a closed-form expression as well and it does not involve back-propogation. The average time per epoch for full-precision DNN and Prox-SGD is roughly the same: 13.06s and 12.21s. So doubling the problem parameters in Prox-SGD does not seem to increase the computational complexity.
>
> These observations are also included in the revised paper.
>
> ***
>
> Comment: The main theoretical result is presented with a sketch of a proof, and I did not attempt to reconstruct the argument from the named sources. It could be useful to provide a full proof (perhaps in an appendix) to allow the work to be self-contained.
>
> Reply: We thank the reviewer for the constructive comment. We have included the full proof to make the paper self-contained.
>
> ***
>
> Last but not least, we thank the reviewer for pointing out the typos. We have corrected them and proofread the whole paper again.
>
> ***
>
> We thank the reviewer again for the constructive comments. We kindly and respectfully ask the reviewer to consider updating the rating if the comments are addressed to the reviewer’s satisfaction.

---

### Author Response · Authors · 2019-11-15
**We sincerely appreciate the reviewers' comments**

The comments have provided a good opportunity to clarify the confusions and greatly improved the quality of our work!

---

### Public Comment · ~Shih-Kang_Chao1 · 2019-12-20
**Selecting mu for sparse NN**

I would like to start by congratulating the authors on the acceptance of a thought-provoking work!

I have a question on the selection of mu for sparse NN. In the numerical experiment, mu = 5*10^(-5). However, there is no sensitivity analysis for the selection of this parameter, like Figure 3 for the epsilon_0. I wonder if the algorithm is sensitive to the selection of mu? In addition, what is the strategy for selecting mu in practice? I ask because the testing accuracy reported in the numerical analysis section does not seem to be the state-of-the-art for, e.g. CIFAR-100 under DenseNet-201 (Fig. 2, central panel), for which the testing accuracy can be 80% plus.

As I have great interest in this work, I would greatly appreciate if there is a chance that my questions can be clarified. Again, congratulations on the achievement!

---

> ### Author Response · Authors · 2020-01-08
> **mu is a hyperparameter, but the performance is not sensitive to its choice.**
>
>
> Thank you very much for the comment!
>
> A numerical test on the sensitivity of the choice of mu was included in the initial submission, but removed in the revised version.
>
> In our current work, mu is a hyperparameter that requires some tuning, because the demand on accuracy and sparsity may vary for different problems. But it is enough to find a coarse range, and within that range, the performance is quite similar.
>
> We did not intend to push for the best score in our simulations. Instead, we are more interested in demonstrating the sparsity achieved by using L1 regularization. Besides, in our simulation setup, the achieved accuracy of ProxSGD is similar to or better than SGD (with momentum) and other adaptive stochastic algorithms. So there is still room for improvement in accuracy.
>
> Thank you again for your comment!
>
> Yang

---

### Decision · Program_Chairs · 2019-12-19

**Decision:**

Accept (Poster)

**Comment:**

This paper proposes a new gradient-based stochastic optimization algorithm by adapting theory for proximal algorithms to the non-convex setting.

The majority of reviewers voted for accept. The authors are encouraged to revise with respect to reviewer comments.